# A Graphical Framework for Knowledge Exchange between Humans and Neural Networks

## Abstract

How could humans better teach, understand, and communicate with artificial neural networks, to correct some mistakes and to learn new knowledge? Currently, network reasoning is mostly opaque. Attempts at modifying it are usually through costly addition of new labeled data and retraining, with no guarantee that the desired improvement will be achieved. Here, we develop a framework that allows humans to understand the reasoning logic of a network easily and intuitively, in graphical form. We provide means for humans to leverage their broader contextual knowledge, common sense, and causal inference abilities: they simply inspect and modify the graph as needed, to correct any underlying flawed network reasoning. We then automatically merge and distill the modified knowledge back into the original network. The improved network can exactly replace the original, but performs better thanks to human teaching. We show viability of the approach on large-scale image classification and zero-shot learning tasks.

## 1 Introduction

The current predominant approach to supervised teaching of neural networks resembles the repetitive act of "cramming" for an exam by repeatedly rehearsing content from flash cards (annotated training datasets). This often positions humans as peripheral to neural network learning, emphasizing data collection and model tuning over possibly deeper interactions. This, in turn, hinders direct and efficient knowledge exchange, making it challenging for humans to convey their knowledge to neural networks and vice versa (Bostrom & Yudkowsky, 2018). As Socrates famously said, "I cannot teach anybody anything. I can only make them think" (Babbitt et al., 1927). This naturally prompts us to explore a new role for humans in teaching neural networks, not only by providing large amounts of annotated data, but also by guiding and refining the network's thought process or reasoning logic through iterative dialogues with human mentors. Here, we hence define a new paradigm for improving the efficiency of human-neural network interaction, emphasizing the need for a better common language (Lake et al., 2017) and techniques that enable direct teaching and learning. By bridging the communication gap between humans and neural networks, we can unlock the full potential of this knowledge exchange; while our work remains a simplified form of a Socratic conversation or dialogue between humans and networks, it is a significant step beyond the prevailing cramming of annotated datasets.

As machine learning (ML) systems become ubiquitous in our lives (Shen et al., 2017; Davenport & Kalakota, 2019; Jumper et al., 2021; Rombach et al., 2022; Kuang et al., 2025), the need to understand, explain, and trust ML systems is ever growing (Zerilli et al., 2019; Vold, 2024). Both humans and neural networks have their respective strengths and can offer valuable insights to one another (Hinton, 2018; Daneshvar et al., 2024). Especially when a network makes mistakes, a good human teacher may be able to contribute additional prior and domain expertise knowledge, causal inference abilities, and common sense, to help correct these mistakes. However, the lack of an effective knowledge exchange interface has made it hard for a human to locate the reason for a network's error, not to mention correct the error.

There are two main challenges for the interaction between humans and networks: (1) Interpretability (Gilpin et al., 2018; Nematzadeh et al., 2025) for humans, i.e., how to understand the reasoning logic of a network

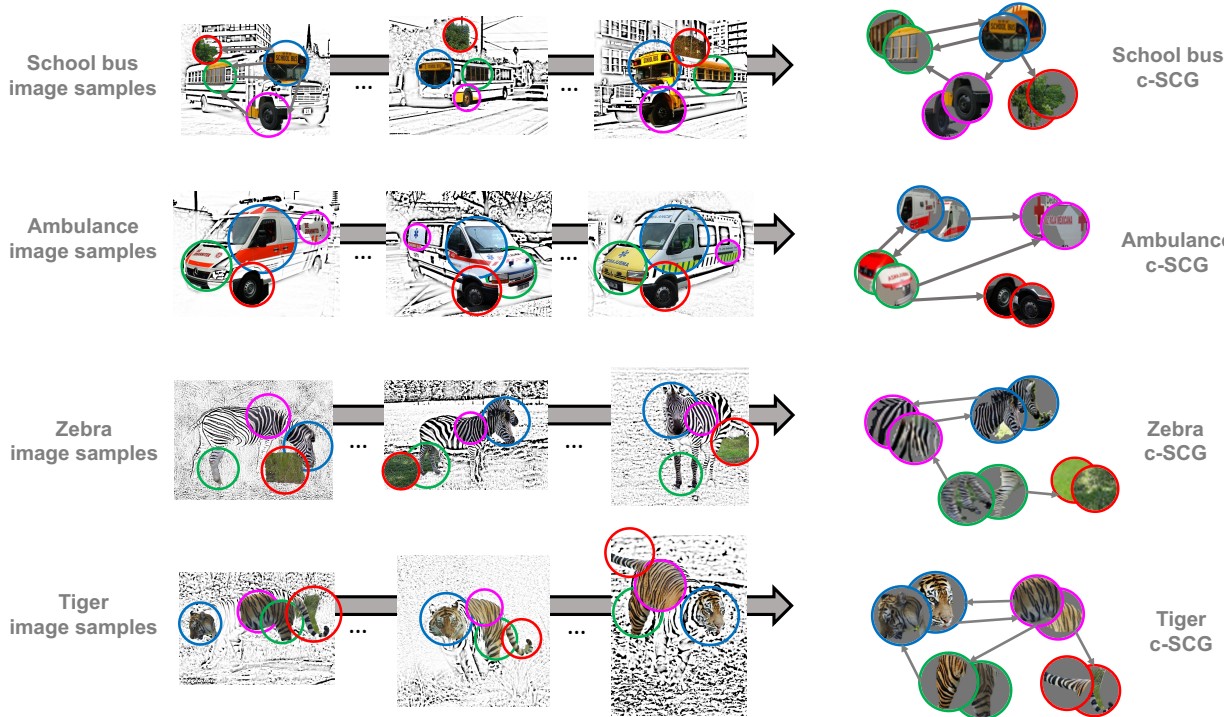

Figure 1: Network-to-human path in our approach shows the reasoning logic of a network to a human, using a Structural Concept Graph as "language". The network used was GoogleNet (Szegedy et al., 2015) trained on ILSVRC2012 (ImageNet) (Deng et al., 2009). Four examples of object classes are shown (one per row). In each one, we highlight the four most important visual concepts according to the original network (different colors), in three instance images. These visual concepts are the ones that most influence the decision of the original network, as discovered through an automated analysis of the network (visual concept extractor). Aggregating these instance-level concepts produces a class-level structural concept graph (c-SCG; rightmost column) for each object class, which captures the most discriminative visual concepts or parts for that class, according to the original neural network, as well as their relationships. Sometimes, the most important concepts for the original network are wrong, possibly because of spurious correlations in the training data, or for some other reasons detailed below (e.g., red circle on top row is a patch of background foliage, which may have often appeared next to school buses during training, but actually is not part of school bus; likewise for a patch of grass with Zebra).

and how to correctly locate the reasons for errors (Ge et al., 2021). (2) Changing a network's logic and decision, i.e., once humans locate an error of the network, how to correct it and improve the network's performance.

Interpreting networks has traditionally been approached through "feature-based" explanation methods (Ghorbani et al., 2019b; Fel et al., 2023) that involve modifying input features (e.g., pixels, super-pixels, word vectors) by either removing (Ribeiro et al., 2016) or perturbing some of them (through zeroing-out, blurring, shuffling) (Sundararajan et al., 2017; Smilkov et al., 2017; Li et al., 2024) to observe the consequences on a network's output. These methods aim to titrate the importance of each feature in the model's predictions. For visual interpretation, class-discriminative attention maps can be generated to highlight image regions that strongly support the network's decision (Zhou et al., 2016; Selvaraju et al., 2017). However, these approaches have been criticized for reliability issues (Ghorbani et al., 2019a; Adebayo et al., 2018), vulnerability to adversarial perturbations (Kindermans et al., 2019), and susceptibility to human confirmation biases (Kim et al., 2018). Additionally, they do not necessarily improve human understanding of the model (Poursabzi-Sangdeh et al., 2021; Salimzadeh et al., 2024).

Interacting with neural networks has been a topic of interest in Human-In-the-Loop Machine Learning (HILML) (Dellermann et al., 2021; Vats et al., 2024; Shen et al., 2024) and Interactive Machine Learning (IML) (Ware et al., 2002). These ML-centered approaches typically involve a pipeline where models are retrained using human-curated data (Dellermann et al., 2021). In this process, humans essentially act as "servers" around the ML process, by being involved in data production, ML modeling, and model evaluation and refinement (Maadi et al., 2021; Colin et al., 2022). However, this limited role constrains human involvement and the potential for domain expert contributions.

Recent research has explored more human-friendly interactions between humans and neural networks. "Tell me where to look" (Li et al., 2018) employs an explainable attention map to rectify segmentation errors in networks, but this approach is restricted to pixel-level masks. Revising Neuro-Symbolic (Stammer et al., 2021) attempts to use explanations as feedback to correct errors or biases in the original network with human intervention. This method requires humans to examine each image to identify potential issues and engage in dataset-specific model training, leading to high costs and low effectiveness. Furthermore, the approach lacks generalization to real-world datasets and does not consider spatial relationships. User interaction has also been introduced in image generation tasks. For instance, Interactive Image Generation (Mittal et al., 2019) can repeatedly modify images based on modifications to the scene graph while keeping the contents generated over previous steps.

Interacting with Large Language Models (LLM) has become a new trend of interest in the field of Explainable Artificial Intelligence (XAI) recently. Benchmark-setting models like OpenAI's GPT series and Meta's Llama series (Touvron et al., 2023) have revolutionized various machine learning tasks, including Human-In-the-Loop Machine Learning. HuntGPT utilizes the capabilities of LLM and XAI to enhance network anomaly detection. Additionally, x-[plAIn] (Mavrepis et al., 2024) introduced an innovative model to make XAI more broadly accessible through a custom Large Language Model created with the ChatGPT Builder. LoRA (Hu et al., 2021) can tune LLMs efficiently but is still driven by a dataset. Furthermore, cross-domain integration requires multiple trained LoRAs to merge various concepts into one domain, and the training time and storage cost are still challenges. Methods like VeRA (Kopiczko et al., 2023) can reduce the number of trainable parameters. LoRA-Composer (Yang et al., 2024) proposes a training-free framework designed for seamlessly integrating multiple LoRAs. Even with these methods, the dataset and training time are still a significant cost for training individual networks. It is critical to address the practical aspects of implementing such systems, particularly regarding prompt engineering costs and the scalability of models.

To address these challenges, we introduce the Human-Neural Network Interface (HNI) for knowledge exchange, offering the following key contributions:

1. HNI employs high-level class-specific visual concepts and their relationships to construct a class-specific structural concept graph (c-SCG) for each class of interest in an image classification task (Fig. 1). The c-SCG represents the key components (concepts; graph nodes) of an object class and their spatial relationships (graph edges). This allows both humans and networks to understand each other using c-SCG as a common "language" for communication, interaction, and knowledge exchange.

2. Through the network-to-human path, the network can utilize c-SCG to present its reasoning logic in a manner that is easily comprehensible to humans. Along the human-to-network path, humans can analyze the network's reasoning logic (c-SCG) and can modify it with their prior knowledge. HNI then employs a Graph Reasoning Network and partial knowledge distillation to transfer knowledge from humans back to the original network, enabling the network to acquire new knowledge from human input.

3. By creating new c-SCGs or modifying existing ones, humans can teach the network to recognize previously unseen objects, thereby establishing a novel pipeline for zero-shot learning.

4. HNI only modifies the original model's parameters with minimal overhead. This design ensures high compute efficiency and avoids the large-scale training or storage costs often associated with LLM-based solutions.

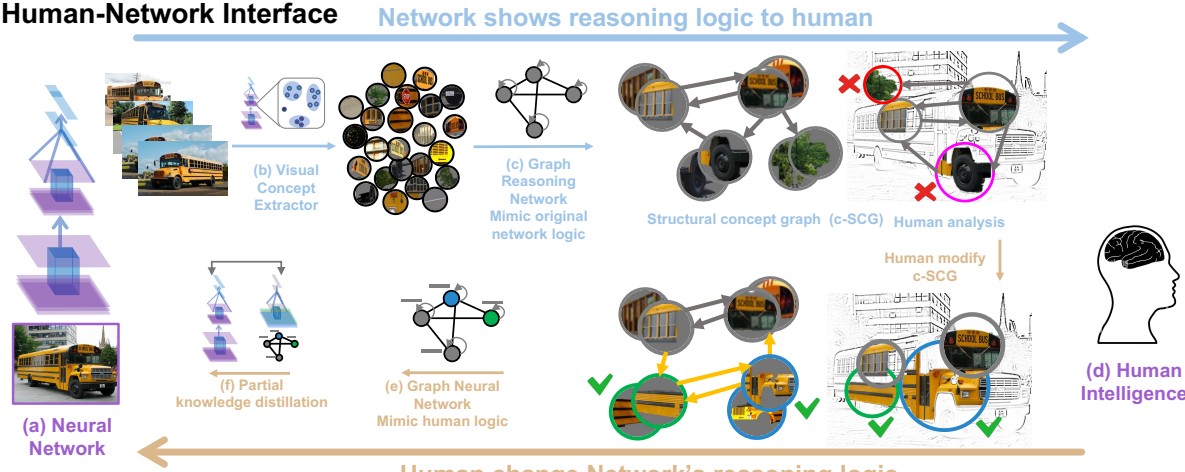

Figure 2: Pipeline for the proposed Human-Network Interface. Top arrow represents the network-to-human path, which shows the reasoning logic of the original network (a) to a human, using structural concept graphs (SCG) as a language. It consists of a Visual Concept Extractor (b), which discovers the most important visual concepts for the network, and a Graph reasoning Network (GRN; c), which aggregates and summarizes concepts and their relationships from many training images into a single class-level structural concept graph (c-SCG) for each class. Bottom arrow represents the Human-to-network path, which changes the network's decision making through human intervention. This is made easy and intuitive by allowing humans to interact with the c-SCG. This path consists of three steps: (1) humans (d) inspect and possibly change a given c-SCG, using their common sense, domain knowledge, and understanding of how spurious correlations may cause errors, to fix errors in the c-SCG. For example, at top-right, tree foliage was used by the original network to recognize school buses, but this is likely a spurious correlation in the training set (many school buses were shown in front of trees); conversely, a wheel was used by the original network, but is not ideal because it is not discriminative of other wheeled vehicles. Humans can choose to substitute these visual concepts with others from the pool extracted by the Visual Concept Extractor, initially ranked less important by the network. Humans can also modify the edges of the c-SCG, to add, remove, or correct relationships between visual concepts. (2) The framework then trains a new GRN with human logic (e), and (3) transfers human knowledge to the original network by partial knowledge distillation (f). The revised network has exactly the same structure as the original, but its weights have been modified following the human interaction. We show in our results that this pipeline is effective at rapidly (in terms of human effort) and interactively correcting network mistakes.

## 2 Related Work

### 2.1 Human-AI Interaction

Human-AI Interaction for Machine Learning (ML) applications aims to best combine human domain expertise and computational power of ML. To satisfy this need, Human-In-the-Loop ML (HILML) (Dellermann et al., 2021; Natarajan et al., 2025) and Interactive Machine Learning (IML) (Ware et al., 2002) have recently emerged. However, both are ML-centered methods which let humans play a "server" role around the ML process, from data production, ML modeling, to model evaluation and refinement (Maadi et al., 2021). This limits human involvement and domain expert performance. "Tell me where to look" (Li et al., 2018) uses an explainable attention map to correct segmentation errors of network, which is thus limited to low-level relationships. User interaction was also introduced in the image generation task, Interactive Image Generation (Mittal et al., 2019) can repeatedly modify images based on modifications to the scene graph while keeping the contents generated over previous steps. In our work, we take a step toward an interface

with which human users and network can more efficiently communicate, interact and exchange knowledge between each other, with no dependency on a given list of attributes.

## 2.2 Interpreting Neural Networks

Research on interpretability methods for neural networks (networks) has received more and more attention (Samek et al., 2021; Gantla, 2025). Some try to explain network by visualizing the correlation between each pixel of the input image and the final outputs. For instance, CAM (Zhou et al., 2016) and Grad-CAM (Selvaraju et al., 2017) can generate class-specific attention maps. Differently, several recent works focus on more human-intuitive concept-level explanations. Lapuschkin et al. (2019) employs spectral relevance analysis to uncover diverse problem-solving strategies that may be hidden by conventional performance metrics. Anders et al. (2022) propose a scalable framework, including Class Artifact Compensation, to detect and mitigate "Clever Hans" behaviors due to spurious correlations, while Achtibat et al. (2023) introduce Concept Relevance Propagation, which unifies local and global explanations to address both "where" and "what" questions in model decisions. Binder et al. (2023) critically examine model-randomization–based sanity checks, revealing limitations in distinguishing meaningful attributions from noise. ACE and TCAV (Ghorbani et al., 2019a; Kim et al., 2018) proposed algorithms to extract meaningful concepts from images and then produce an understandable explanation. VRX (Ge et al., 2021) uses visual concepts to explain a network's reasoning logic. While these approaches yield valuable insights into a model's internal reasoning, they do not permit experts to intervene or correct that logic, leaving no direct way to align human expertise with the network's reasoning.

Recent "concept-based" research provides explanations in the form of high-level human concepts (Zhou et al., 2018; Ghorbani et al., 2019a;b; Lang et al., 2021). These methods focus on extracting or revealing important visual concepts, rather than pixels or features, to explain the original model. In addition, "graph-based" methods have been explored to interpret networks by learning explanatory graphs that reveal the knowledge hierarchy hidden inside pre-trained CNNs (Zhang et al., 2018; 2020) and constructing neural prototype trees for interpretable fine-grained image recognition (Nauta et al., 2021). However, these approaches do not clarify the network's reasoning logic or elucidate how spatial relationships and interactions among image regions or concepts may affect decisions. The recently proposed visual reasoning explanation (Ge et al., 2021) mimics the original network's reasoning logic and provides logical, easy-to-understand explanations for final decisions, but it cannot directly influence the network's performance to achieve closed-loop interaction. X-NeSyL(Díaz-Rodríguez et al., 2022) fuses deep learning representations with expert knowledge graphs to improve both performance and explainability. However, this approach requires extensive domain expertise to construct accurate and comprehensive knowledge graphs, which can be a significant limitation in practice.

## 2.3 Human Alignment in Cognitive Science

Recent research has increasingly emphasized the importance of aligning neural network representations with human cognitive structures. Muttenthaler et al. (2023) shows that while model scale and architecture play a minor role, the training dataset and objective function are critical in shaping how well network representations mirror human similarity judgments—highlighting that certain human concepts (e.g., food and animals) are naturally well-represented, whereas others (such as royal or sports-related objects) are not. Complementing these insights, Muttenthaler et al. (2023) proposes a unifying framework that bridges these fields and identifies open challenges in measuring and improving alignment. In an applied context, Demircan et al. (2024) evaluates the alignment between human learning trajectories and neural network representations in image-based tasks, finding that factors such as training dataset size and contrastive multimodal training are crucial for models to generalize in a human-like manner. Yang & Zou (2021) proposes a graph-based interactive reasoning approach for human-object interaction detection and illustrates how leveraging interactive semantic graphs can capture complex relationships akin to human reasoning, thereby furthering the goal of human alignment in AI. In our work, we take a step toward an interface with which human users and network can more efficiently communicate, interact and exchange knowledge between each other, with no dependency on a given list of attributes.

### 2.4 Graph Neural Networks

Graph Neural Networks (GNN) are deep learning methods that operate on graph domains, which learn to represent graph nodes, edges, or subgraphs by low-dimensional vectors (Zhou et al., 2021). Considering the trackable information-communication properties of GNNs, many reasoning tasks have adopted GNNs as a tool, such as VQA (Teney et al., 2017; Norcliffe-Brown et al., 2018), scene understanding (Li et al., 2017), and semantic explanations (Ge et al., 2021). In this work, we use a GNN-based Graph Reasoning Network (GRN) as the bridge between networks and humans. It takes SCG as input and transfers knowledge with network through knowledge distillation.

### 2.5 Zero-shot Learning

Zero-shot learning aims to train a classifier that can classify testing instances that belong to classes that were never seen before. Existing works include using the one-vs-rest solution (Verma & Rai, 2017), synthesizing pseudo instances of the unseen class (Guo et al., 2017), projecting feature space instances and semantic space prototypes into a common space (Palatucci et al., 2009), and using similar seen class as the positive instance of the unseen class (Gan et al., 2016). Huynh & Elhamifar (2020) conduct compositional zero-shot learning by using a feature composition framework to extract and combine features of attributes to construct fine-grained attributes for unseen classes. Jia et al. (2021) uses active zero-shot learning which promotes human-AI teaming by actively modifying the class-attribute matrix. However such attribute labels may not always be available in real-world scenarios. Here, we propose a new pipeline with HNI, which makes no assumption on the availability of attribute label, humans help build the understanding of the new class using learned visual concepts and structure. Then HNI transfers the knowledge of the unseen class back to the network.

## 3 Methods

Our proposed Human-network Interface (HNI) to bridge the interaction between human and neural networks is visually summarized in Fig. 2. There are two main paths.

### 3.1 Network-to-Human

The network-to-human path explains the network's reasoning logic, the understanding of the network for each class, represented as class-specific Structural Concept Graph (c-SCG). Each c-SCG is bound to one class (Fig. 2 shows the c-SCG of school bus), where the nodes represent the important visual concepts that the original network considered most important in identifying the class of interest, and edges represent the pairwise structural relationships (dependencies) between concepts. As shown in Fig. 2 (top), given a trained network, there are two main steps to explain the reasoning logic of the network to human users:

(1) Using **Visual Concept Extractor (VCE)** to discover representative visual concepts for each class of interest. The detailed procedure follows (Ge et al., 2021): To discover concepts for each class, we collect 50 to 100 images of the class. We first use top-down gradient attention (Grad-Cam (Selvaraju et al., 2017)) to constrain the relevant regions for concept proposals to the foreground segments, thereby ruling out irrelevant background patterns for this class. Then, we follow a workflow similar to that in the ACE paper (Ghorbani et al., 2019b), which includes image segmentation, feature extraction, and clustering patches in latent space to obtain concept candidates. These candidates are then sorted based on an importance score, akin to the method in (Kim et al., 2018). The primary distinction between our method and ACE lies in our use of Semantic SAM (Li et al., 2023) for segmenting the input images and generating concept candidates. This approach yields better concept candidates due to the rich semantic content provided by Semantic SAM (as shown in Fig. 3 step 1). After that, we obtain the concept pool (each concept is represented by one mean feature vector, sorted by importance score) for each class, which will serve as a source of concept candidates when human users modify concepts (nodes) for c-SCG.

(2) Using **Graph Reasoning Network (GRN)** to mimic the decision-making process of the original network with knowledge distillation. Specifically, we distill the reasoning logic of the original network into

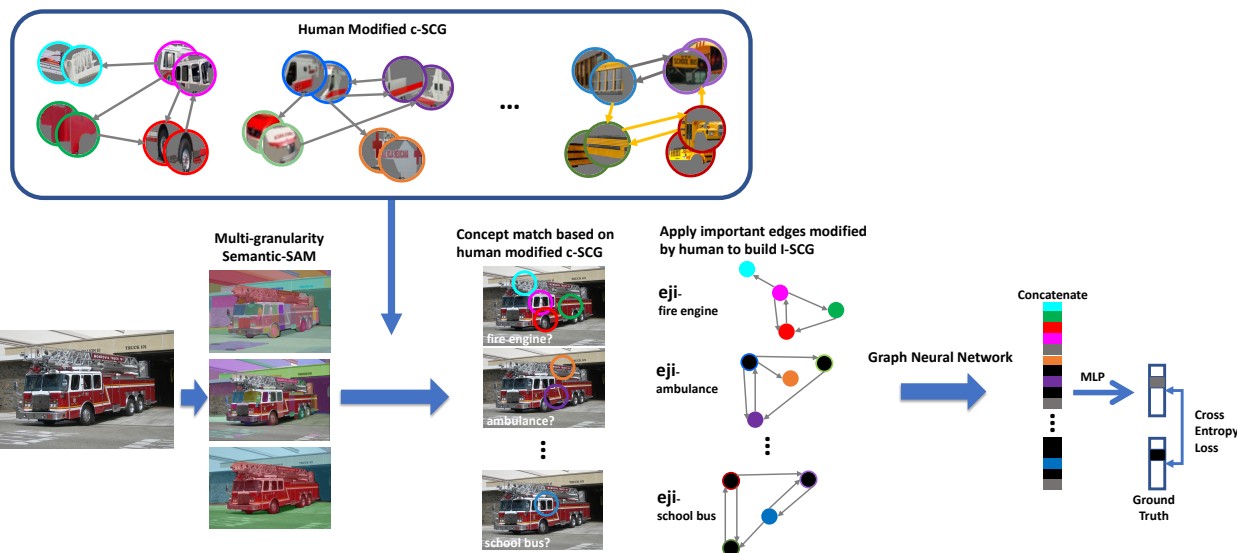

Figure 3: Pipeline of building a c-SCG that reveals the reasoning logic for each class of interest of the original network.

a graph-based network, GRN, which uses more interpretable visual concepts. As a Graph Neural Network (GNN) based network, GRN takes a graph as input but has a similar decision as the original network (CNN). During knowledge distillation, to train GRN based on the built c-SCGs (one for each of $n$ classes of interest), we need to establish connections between training images and the c-SCGs.

To build a c-SCG that reveals the reasoning logic for each class of interest of the original network, we select the top $k$ ($k$=4 in our experiments) important concepts and their mean feature vectors as nodes, as well as edges between them. Fig. 1 shows the c-SCG which represents the understanding of the original network to each class. Each directed edge in an SCG edge$_{ji} = (v_j, v_i)$ has two attributes: 1) representation of the spatial structural relationship between nodes; 2) dependency $e_{ji}$ (a trainable scalar) between concepts $v_i$, $v_j$. The edge features can reveal the importance of interactions between visual concepts that may be crucial for the final decision. c-SCG extracts the relationship between visual concepts during training (next step) and incorporates them as edge features in c-SCG. By showing c-SCG for each class of interest, the network-to-human path provides easy-to-understand insights for human users on the reasoning process of the network, which is also a foundation for the Human-to-network path. The edges (structural relationship and dependency) are fully connected at the beginning, and then, using the learning in the following step 2, we select and only keep the important edges.

After we generated the c-SCG, we create, for each training image $I$, a set of up to $n$ image-level structural concept graphs (I-SCGs). Each I-SCG is computed from both the training image $I$ and one of the $n$ c-SCGs: Given the input image $I$, we use multi-granularity segmentation Semantic SAM to break the image into patches, which become the concept candidates. In the concept matching step (Fig. 3 step 2), for each class of interest $c$, we match features of the segmented patches to the stored anchor representation (i.e., mean feature vectors) of top $k$ concepts deemed important for $c$ using a similarity metric (e.g., Euclidean distance). When at least one of the top $k$ concepts of class $c$ is detected in image $I$, an I-SCG for class $c$ will be constructed based on the template from the c-SCG of class $c$. Here I-SCG uses patch features instead of the concept anchor features as node features and we calculate the edge features based on the spatial relationship between detected concepts in image $I$. This way, we can generate up to $n$ I-SCGs for the input image $I$ considering all $n$ classes of interest. GRN takes the I-SCGs as input, and we use knowledge distillation to transfer the decision-making logic of the original network to GRN. Using SCG to effectively reveal the original network's reasoning logic has been validated with extensive experiments in (Ge et al., 2021), in which the authors evaluated the logical consistency and faithfulness between SCG explanation and the original network.

### 3.2 Human-to-network

The human-to-network path transfers human's knowledge to network, in order to improve the original network's performance and generalizability. There are three main steps: (1) User modifies c-SCG: after understanding the network's reasoning logic with network-to-human path, users can verify whether the decision logic is reasonable or consistent with their understandings. If not, human users are able to actively correct the decision logic by updating the c-SCG (e.g., deleting a visual concept and changing the structural relationship between concepts) efficiently. (2) To represent human-modified logic, we use the modified c-SCG as a template to automatically rebuild I-SCGs for images, and we train a new Graph Reasoning Network (GRN) , with ground truth image labels. (3) To let the original network learn human-modified logic, we propose partial knowledge distillation to transfer the logic of GRN, which has incorporated the knowledge and prior from human users, back to the original network. We describe the three steps in detail in the following subsections.

#### 3.2.1 Human modifies c-SCG

The Network's understanding of any specific class can be shown as a single c-SCG: the nodes (visual concepts) represent the crucial visual evidence or clue for the network to identify this class; the edges encode the structure relationships and dependencies between concepts. After understanding the meaning of c-SCG, human users can intuitively make modifications to c-SCGs (e.g., removing the incorrect nodes/edges) based on their knowledge or other priors, in order to improve the network's performance. There are two main types of modification: nodes and edges, corresponding to changing the visual concepts and the relationships between concepts respectively. Fig. 2 bottom path and Fig. 4 (c1, c2) show examples of human users modifying c-SCG.

**Node (concept) modification:** human users can easily identify non-casual concepts in c-SCG. Fig. 4(c1) shows an example of node modification. In some cases, nodes may be irrelevant to the class-of-interest (e.g., a background object always appears together with the object of interest; e.g., see branch and fire engine), or not representative/unique (e.g., an object part that is common among many classes; see the wheel in different vehicle classes). To substitute these two concepts with more representative and discriminative ones, human users can go back to the concept pool extracted by the VCE in the network-to-human path and select better visual concepts to improve the c-SCG (Fig. 4(c1, c2)).

**Edge (concept relationship) modification:** Edges shown by c-SCG are the important dependencies selected based on the values of $e_{ji}$. Humans can modify them to remove non-stable or independent relationships between concepts. This may happen when substantial biases exist in training, as the network may discover, in the training set, stable and dependent relationships between concepts which in fact do not always hold in real-world scenarios (e.g., the relative position of a cheetah's body and tail in Fig. 4(c2)). Human users can remove this edge on the c-SCG to correct the bias. In practice, modifications of nodes and edges can happen simultaneously to handle more complex situations. Note that **to modify the decision reasoning logic for one class, human users only need to modify the corresponding c-SCG (one graph with $k = 4$ nodes per class)**: e.g., they can substitute one concept with another from the pool, or add/delete edges. They do not need to modify image-level I-SCG for every training image. After updating the c-SCG, our framework automatically applies this modification to all image-level I-SCGs. Human modification of c-SCGs is the first step in the human-to-network path, where our framework provides an intuitive way for human users to convert their knowledge, common sense, and priors into a description in the same language that our framework uses. Next, we will show how the knowledge of human users can be transferred back to the network in the following subsections.

#### 3.2.2 Training Graph Reasoning Network (GRN) with Human's Logic

Typically, the set $S_I$ of classes that require human intervention is a subset of the set of all $S$ classes ($S_I \subset S$). This setting is flexible and efficient: no matter how many classes the original network can predict (e.g., 1,000 classes in ImageNet), users may only want to modify the logic of a small subset of classes in question (e.g., some vehicles are easily confused with each other). In this case, we build a GRN that targets the logics of these classes only, which is more efficient to users. For each class of interest $c \in S_I$, the network-to-human

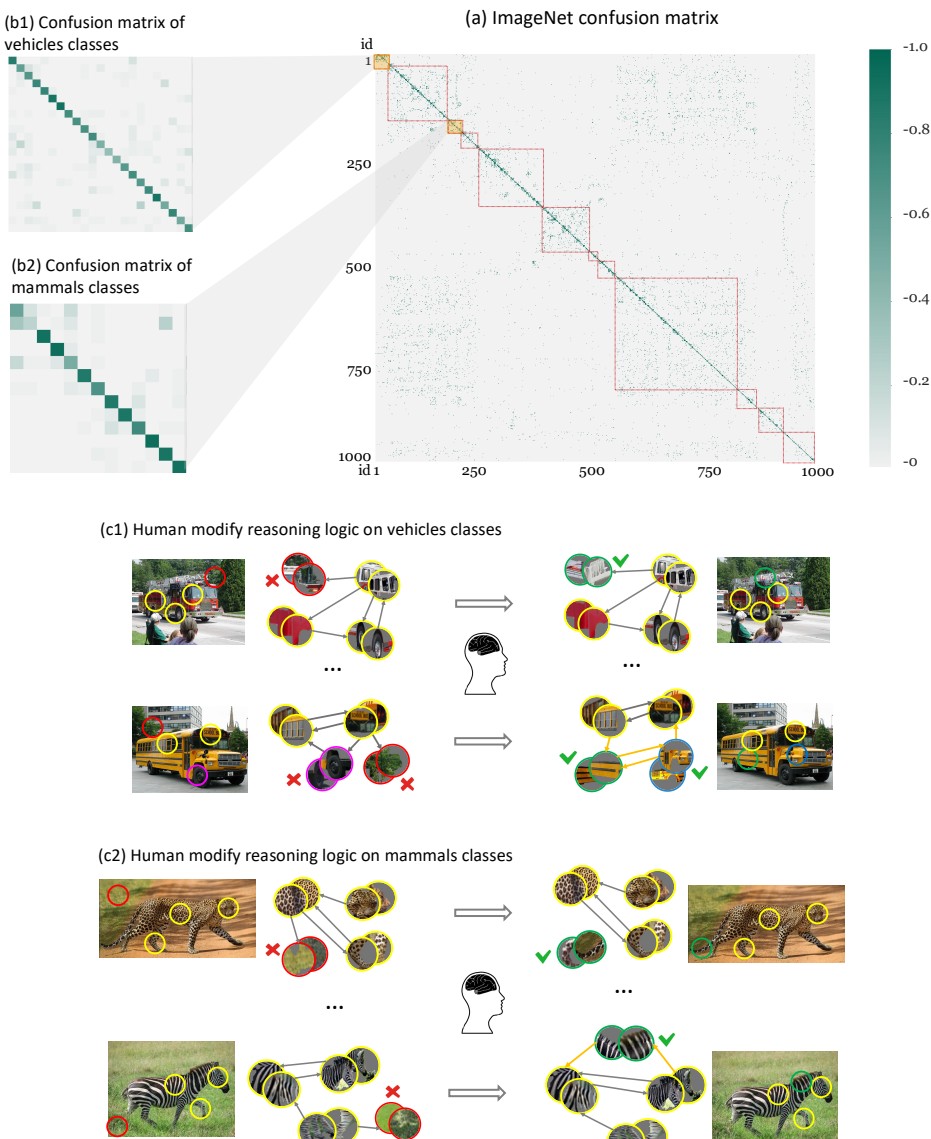

Figure 4: Humans can improve a network's performance with HNI. We conduct large-scale experiments on the ImageNet dataset, which contains 1,000 real-world classes. (a) Confusion matrix of a 1,000-class original GoogleNet image classification network trained on ImageNet. Most of the errors are within each of 12 super-classes that correspond to groups of related classes (e.g., mammals, vehicles, birds, etc). For each chosen class, the network-to-human pass was used to show the reasoning logic of the original network as a c-SCG to a human operator (c1, c2). The operator spotted and corrected any reasoning errors of the network. The human-to-network pass then distilled the human-modified logic back to the original network with the help of graph neural network and partial knowledge distillation.

path reveals its reasoning logic c-$SCG^c_{network}$. After human's analysis and modification, some classes may have updated c-SCGs, c-$SCG^c_H$ after incorporating human user's knowledge. c-SCG as a class representation cannot produce final decisions by itself; hence we reuse the GRN from network-to-Human path to infer a prediction.

Fig. 5 shows the pipeline of training GRN with the c-SCG updated by human users. Given an input image $I$, the first two steps of the processing (Multi-granularity segmentation and Concept match to build I-SCG)

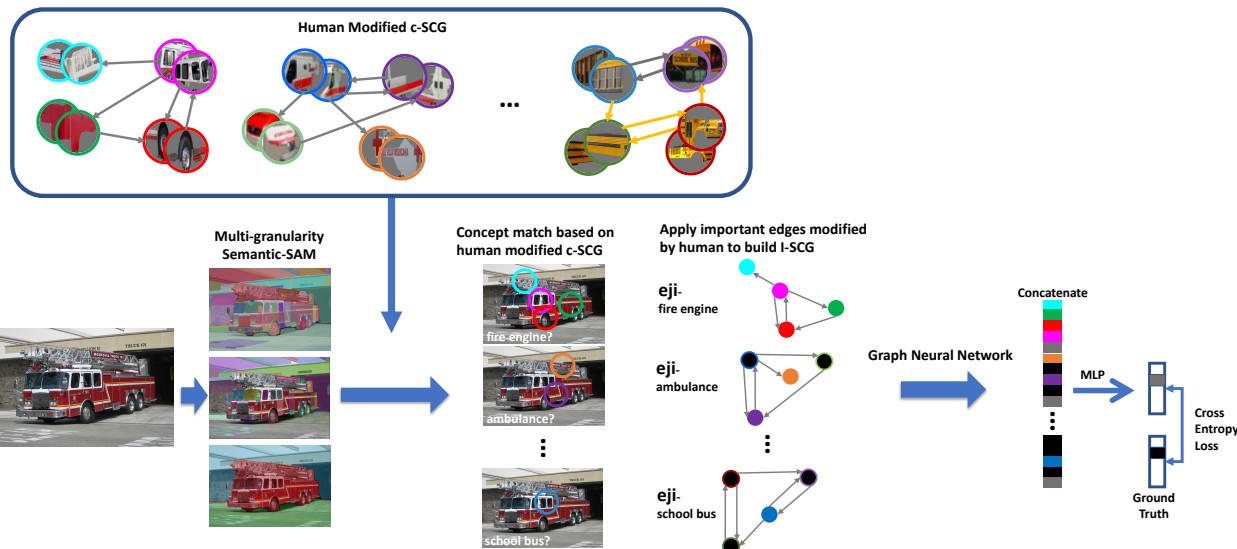

Figure 5: Pipeline of training Graph Reasoning Network with Human modified c-SCG. Given input $I$, we conduct multi-granularity semantic segmentation and concept match based on human-modified c-SCG. In the concept matching step, we attempt to match the c-SCG of each class of interest to the concepts extracted from the current input image. Color circles represent the matched concepts for each class of interest. Black dummy nodes denote undetected concepts. For example, for the input image shown, all concepts for the Fire Engine class were matched, but only 2 concepts of Ambulance could be found in the image, and only 1 concept of School Bus. Subsequently, the GRN aggregates all matched concepts and uses those to support its predictions.

are the same as the GRN training in network-to-human path, while the objective is different. Here our goal is to obtain better performance by incorporating human user knowledge. The matched I-SCGs go through the graph convolution backbone and MLP in GRN and finally predict the image label with cross-entropy loss as the objective function. The trained GRN can then produce decisions based on human-corrected reasoning logic because the input I-SCGs are derived from c-$SCG_H^c$.

### 3.2.3 Transfer Reasoning Logic to Network with Partial Knowledge Distillation

The last step to is to transfer the user-updated reasoning logic and knowledge in GRN back to the original network. To avoid catastrophic forgetting and negative impact on the classification performance of other classes, we developed partial knowledge distillation as a new knowledge transfer method.

Fig. 6 illustrates the process of partial knowledge distillation to transfer human user's knowledge from GRN back to the original network. Modified classes $S_I$ are a subset of all classes $S$. For the set of unmodified classes $S_U = S \setminus S_I$, we want to maintain their reasoning logic while we update those of $S_I$ in the original network. Hence two teacher models provide soft labels together. GRN $Net_{T1}$ provides the probabilities of modified classes; the original network $Net_{T2}$ with *fixed* parameters provides the probabilities of unmodified classes. The student model $Net_S$ shares the same architecture as the original network and is initialized with the weights of the original network. Formally, the overall loss during partial knowledge distillation is as follows:

$$L = \alpha L_{soft} + \beta L_{hard} \tag{1}$$

where $\alpha$ and $\beta$ are the weightings of the two terms during distillation. For the soft label term:

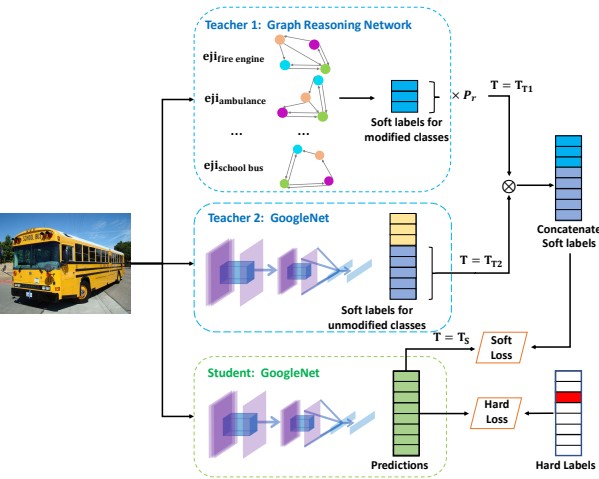

Figure 6: **The pipeline of Partial Knowledge Distillation.** Different from traditional knowledge distillation, partial knowledge distillation adopts two teachers with different expertise: GRN (teacher 1) focuses on classes of interest (6 classes in this example), and the fixed original network (teacher 2) focuses on the remaining classes (the classes we don't want to change, 14 classes in this example). After distillation with different temperatures and concatenation, we can use both soft labels and hard labels to train the student model.

$$L_{soft} = -\sum_{c=1}^{N} \hat{p}_c^{T_T} log(q_c^{T_S}); \qquad q_c^{T_S} = \frac{exp(z_c/T_S)}{\sum_{k=1}^{N} exp(z_k/T_S)} \qquad (2)$$

where $\hat{p}_c^{T_T}$ denotes the probability value of class $c$ in the combined soft label with temperature $T_T$ from two teacher models. $q_c^{T_S}$ denotes the probability value of class $c$ in the student prediction vector with temperature $T_S$. $N = |S|$ denotes the number of classes in the original network. For $q_c^{T_S}$, $z_c$ denotes the logits of $Net_S$, which is the un-normalised predictions. The combined soft label $\hat{p}^{T_T}$ is the combination of two soft labels $p^{T_{T1}}$ and $p^{T_{T2}}$ from two teacher models $Net_{T1}$ and $Net_{T2}$. $p^{T_{T2}}$ is a vector with length $N$, $p^{T_{T2}} \in \mathbb{R}^N$, while $p^{T_{T1}}$ is a vector with length $n$, $p^{T_{T1}} \in \mathbb{R}^n$, where $n = |S_I|$ denotes the number of classes in the GRN, which is also the number of modified classes, $v_c^1$ and $v_c^2$ denote the logits of the teacher models $Net_{T1}$ and $Net_{T2}$ respectively:

$$\hat{p}_c^{T_T} = \begin{cases} p_c^{T_{T1}} Pr, & \{c \in S_I\} \\ p_c^{T_{T2}}, & \{c \in S \setminus S_I\} \end{cases} \quad s.t. \quad p_c^{T_{T1}} = \frac{exp(v_c^1/T_{T1})}{\sum_{k=1}^{n} exp(v_k^1/T_{T1})} \quad p_c^{T_{T2}} = \frac{exp(v_c^2/T_{T2})}{\sum_{k=1}^{N} exp(v_k^2/T_{T2})} \qquad (3)$$

in which $Pr \in (0,1]$. To obtain the combined soft label $\hat{p}_c^{T_T}$, we first compute the sum of the probability of all classes of interest in $p^{T_{T2}}$. $Pr = \sum p_c^{T_{T2}}$ for all $\{c \in S_I\}$, which represents the probability proportion of the $n$ modified classes w.r.t. all classes $N$ in the original network $Net_{T2}$. We then replace the value of each class of interest in $p^{T_{T2}}$ with the scaled value in $p^{T_{T1}}$ to form the combined soft label. The prediction of teachers may be erroneous, and we use ground-truth labels as hard labels to provide stronger constraints to $Net_S$, correcting these errors from teacher models.

$$L_{hard} = -\sum_{c=1}^{N} g_c log(q_c^1) \qquad (4)$$

where $g_c$ denotes the ground truth label for class $c$, $q_c^1$ is the probability value of class $c$ in the student prediction vector under temperature 1. Eq. 1 transfers knowledge from GRN back to the original network.

To summarize, we use Graph Reasoning Network (GRN) in both the network-to-human path and human-to-network path respectively (Fig. 2) with the same structure. In the network-to-human path, GRN simulates the reasoning logic of the original network with knowledge distillation. In the human-to-network path, after human users modify the c-SCG for classes of interest, GRN uses the modified c-SCG to automatically derive I-SCGs for each image which will be used to train the GRN using ground truth labels (Fig. 3). Once the training of GRN is done, we transfer the knowledge of GRN back to the original network, for which we proposed partial knowledge distillation, where the newly-trained GRN becomes the teacher model to train the original network using GRN's outputs as soft labels (Fig. 6). We summarize the input/output details of the HNI pipeline and each module/process in Table 1.

Table 1: Input and output summarization of the whole HNI pipeline and each module/process

| Module / Process / Pipeline | Input / Collaborator | Output |
|---|---|---|
| Human network Interface (HNI) Pipeline | Original network | Modified network |
| 1 network-to-Human path | Original network | c-SCG, Reasoning logic explanation |
| 1.1 Visual Concept Extractor (VCE) | Original network, 50 to 100 images for each class | Important visual concepts for each class |
| 1.2 Image-level SCG (I-SCG) building | Original network, images, visual concepts | I-SCG |
| 1.3 Graph Reasoning Network (GRN) | Original network, Image-level SCG (I-SCG) | GRN (mimic original network), c-SCG |
| 2 Human-to-network path | Original network, class-specific SCG (c-SCG) | Modified network |
| 2.1 Human involved logic modification | c-SCGs | human-modified c-SCG |
| 2.2 GRN independent training | I-SCG built with human-modified c-SCG | GRN with human reasoning logic |
| 2.3 Partial knowledge distillation | Original network ($Net_S$), GRN ($Net_{T1}$), Original network (fix)($Net_{T2}$), training images | Modified network |

# 4 Experiments

## 4.1 Implementation details

### 4.1.1 Human experiment

Human experiments used a web-based interface to show the original reasoning of the network (c-SCG with the top 4 visual concepts) and also an additional 21 top-ranked visual concepts. Participants were first asked to select the best 4 visual concepts out of the total pool of 25. They were instructed that in some cases, the original 4 concepts chosen by the network may already be the best ones, but in other cases they might contain some errors (e.g., patch of grass for zebra class as shown in Fig. 1). After selecting concepts, participants were shown the edges in the original c-SCG and were asked whether they wanted to modify them to change the importance of relationships between concepts (additional details in Appendix).

### 4.1.2 Concept matching

In the multi-resolution segmentation step, to extract features for each patch resulting from image multi-resolution segmentation, we resize the patch and use the original network to compute features after a specific layer, e.g., "layer4.1.conv2" layer of ResNet-18. For each discovered concept, we store the mean vector of all patches belonging to this concept as an anchor for future concept matching given any image.

In the concept match step (same as Fig. 3 step 2), for each class of interest $c$, we match candidate features to the stored anchors (mean concept vectors) of top $k$ concepts in the concept pool, and we construct an I-SCG for image $I$ and class $c$, based on similarity (Euclidean distance) between image patch feature and concept anchor feature. Specifically, if the nearest image patch regards the Euclidean distance between the concept anchor feature and the image patch feature is smaller than a threshold $t$, then we will identify this patch as a detected concept. Otherwise, we will use dummy nodes (all feature values equal to a small constant $\epsilon$) to represent that undetected concept.

We empirically choose the threshold $t$ from observation when matching concepts mean vector to patches segmented from images and form image-level SCG (I-SCG). Specifically, the distance in the positive match and negative match have different orders of magnitudes in latent space, and the performance is not sensitive to the selection of $t$. We plan to explore more in this direction as to future work.

### 4.1.3 Graph Reasoning Network

The network architectures of Graph Reasoning Network ($GRN$) are shown in Table. 2. $GRN$ consists of two parts, Graph Neural Network $G$ and Embedding Network $E$. $G$ takes $n$ hypotheses $\mathbf{h} = \{h_1, h_2, ...h_n\}$ (each hypothesis $h_i$ is in the form of Structural Concept Graph (SCG), and each SCG has $m_n$ nodes as well as $m_e$ edges) as input, and output $n$ feature vectors ($G(h_i)$) which concatenate all updated node and edge features of $h_i$. In $G$, we use class-specific $e_{ji}^c$ for different hypotheses in each GraphConv layer. $E$ concatenates all $n$ feature vectors from all the hypotheses into a long vector and maps it into $n$ dimensional vector ($1 \times n$) with a 4 layer MLP, where $n$ is the number of classes of interest. "node" denotes node feature, "edge" denotes edge feature, "GraphConv" is graph convolutional layer, "ReLU" denotes ReLU activation function, "BN" denotes batch normalization, and "FC" denotes fully connected layer.

Table 2: Network architectures of Graph Reasoning Network $GRN$. ($l$ is the length of node feature, $m_n$ is the number of nodes in each SCG, $m_e$ is the number of edges in each SCG, and $n$ is the number of classes of interest)

| Part | Input $\to$ Output Shape | Layer Information |
|------|--------------------------|-------------------|
| $G$ | node:$(l \to 64)$; edge:$(4 \to 5)$ | GC-$(e^c_{ji})$, ReLU, BN |
| | node:$(64 \to 32)$; edge:$(5 \to 5)$ | GC-$(e^c_{ji})$, ReLU, BN |
| | node:$(32 \to 32)$; edge:$(5 \to 5)$ | GC-$(e^c_{ji})$, ReLU, BN |
| | node:$(32 \to 32)$; edge:$(5 \to 5)$ | GC-$(e^c_{ji})$, ReLU, BN |
| | node:$(32 \to 32)$; edge:$(5 \to 5)$ | GC-$(e^c_{ji})$, ReLU, BN |
| $E$ | $((32m_n + 5m_e) \times n) \to (128)$ | FC-$((32m_n + 5m_e) \times n, 128)$ |
| | $(128) \to (64)$ | FC-$(128, 64)$ |
| | $(64) \to (32)$ | FC-$(64, 32)$ |
| | $(32) \to (n)$ | FC-$(32, n)$ |

#### 4.1.4 Partial knowledge distillation

For all partial knowledge distillation experiments in the Human-to-network path, we use SGD with initial learning rate = 0.1, and it will multiply 0.1 every 30 epochs. Batch size = 4, the coefficient of $L_{soft}$, $\alpha = 2.5$, the coefficient of $L_{hard}$, $\beta = 1$. For Nodes (concepts) modification experiment , we use $T_s = 1, T_{T1} = 1.5, T_{T2} = 1$, For edges (concepts relationship) modification (4.1.2 in the main paper), we use $T_s = 2, T_{T1} = 2$. For zero-shot learning experiments (4.2 in the main paper), we use $T_s = 1.5, T_{T1} = 1.5$. We use knowledge distillation to train original model for 100 epochs.

#### 4.1.5 Compute Resources

We use 2 RTX-2080 and 2 Telsa V100. We train knowledge distillation on RTX-2080, which costs around 2 hours for 100 epochs on RTX-2080, and we train GRN on Telsa V100, which costs 1 hour for 100 epochs.

Table 3: Human improves a network's performance with HNI: experiments on six different image classification tasks (performance is tabulated as percent correct classification).

| Datasets | # images | Classes | # classes | original performance | performance with HNI |
|----------|----------|---------|-----------|----------------------|----------------------|
| Cats | 2382 | modified classes | 2 | 88.33 | 91.64 (+3.31) |
| | | all classes | 12 | 93.06 | 93.61 (+0.55) |
| Cars | 8144 | modified classes | 2 | 83.33 | 91.67 (+8.34) |
| | | all classes | 10 | 86.33 | 88.33 (+2.00) |
| Monkeys | 1642 | modified classes | 3 | 78.75 | 85.00 (+6.25) |
| | | all classes | 10 | 90.00 | 93.61 (+3.61) |
| Flowers | 22267 | modified classes | 3 | 78.94 | 85.76 (+6.82) |
| | | all classes | 10 | 82.37 | 85.57 (+3.2) |
| Fashion | 16186 | modified classes | 3 | 61.71 | 67.14 (+5.43) |
| | | all classes | 6 | 73.79 | 74.37 (+0.58) |
| Buildings | 5063 | modified classes | 3 | 47.78 | 57.78 (+10.00) |
| | | all classes | 17 | 53.73 | 54.51 (+0.78) |

### 4.2 Results

#### 4.2.1 Experimental results on six image classification tasks

We first evaluate the performance of HNI on six different tasks: Cats classification Parkhi et al. (2012), Cars classification Krause et al. (2013), Monkeys classificationRenard et al. (2018), Flowers classification Saxena (2022), Fashion products classification Xiao et al. (2017) and Buildings classification Philbin et al. (2007). The model was first trained on the original dataset for each task. We could then inspect the confusion matrix and find the most confused classes as the interest classes for which we could modify the logic. The network-to-human path showed the reasoning logic for those interest classes with structural concept graphs.

Then, humans conducted modification on interest classes and the human-to-network path transferred the human reasoning logic back to the original model.

Specifically, in each dataset, an inspection of the confusion matrix after the initial network training revealed the most confused classes (e.g., for Cars classification, it was convertible vs. coupe; for building, it was Cornmarket, Hertford, and Oxford; for Flowers classification, it was Orchid, Lily, and Iris). We then asked humans to edit the corresponding c-SCGs: substitute a visual concept in each graph with another promising concept from the concept pool, or modify the structural relationship between concepts. It is important to note that humans only need to edit one graph per class, and the modifications will automatically propagate to every instance image when transferring the modified logic back to the original network via the human-to-network path. Humans do not need to modify each image in the dataset individually. For example, for Cars, humans only modified two graphs with 4 nodes each: one for convertible and one for coupe.

Table 3 shows the performance before and after humans modified the reasoning logic through HNI to improve the original network's performance. Human modification improved classification accuracy for all modified classes (+3.31% to +10%). In addition, this process did not decrease the overall accuracy of the network on all classes. Overall, with typically $\sim 1$ minute of human work per class to inspect and possibly correct one c-SCG per class, a significantly improved network was created that is a direct drop-in replacement for the original network (exactly identical structure as the original).

Table 4: Humans can improve a network's performance with HNI. We conduct large- scale experiments on the ImageNet dataset, which contains 1,000 real-world classes. We show results of one of the large-scale experiments, for the superclasses of mammals (13 classes, 13,000 images), to show how one can use HNI to improve performance within a superclass without degrading performance of other classes.

| Classes | # classes | original Accuracy | Accuracy with HNI |
|---|---|---|---|
| modified classes | 13 | 88.33 | 91.64 (+3.31) |
| all classes | 1000 | 93.06 | 93.61 |

### 4.2.2 Experimental results on ImageNet classification tasks

We then evaluate the performance of an ImageNet pretrained GoogleNet classifier Szegedy et al. (2015) on the validation dataset of ImageNet. Fig. 4(a) shows the confusion matrix (the misclassified samples). For visualization, we re-order the class sequence based on the class hierarchy proposed in Bilal et al. (2017). Specifically, the classes in the same red squares belong to the same super-class; for instance, vehicles, mammals, birds, etc. We use two experiments (the local logic of mammals and vehicles) to show how humans can improve the performance of network by modifying the reasoning logic. Most of the time, we do not need to modify the reasoning logic of all classes, but only of classes of interest, which form a local logic. For instance, as different classes of mammals are easily confused, we asked human users to help improve the mammal classification performance (local logic) with our framework (Fig. 4(b2) shows the confusion matrix of 13 mammal classes). The goal is to improve the performance of classes of interest (13 classes of mammals), while avoiding performance degradation of other classes (Fig. 4(c2)). We first use the network-to-Human path of HNI to visualize the reasoning logic of the original network. A human user can modify the concepts in question. Table 4 shows the c-SCG comparison before and after modification. In this first large-scale experiment, we engaged experts with a background in machine learning to modify the reasoning logic. We then trained the GRN with the updated c-SCG (details in method section). To transfer human logic back to the original network, we uses partial knowledge distillation (Methods). We can see that the accuracy of mammals improved +2% and the overall performance improved a little. The experimental results shows that humans can accurately modify the reasoning logic of the interest class to improve the performance of the original network.

We then conducted a similar experiment on the local logic of vehicles. The test results are shown in Fig. 7. We asked experts with a background in machine learning to modify the reasoning logic. As shown in Fig. 7(e1), the accuracy on vehicles improved +3.5% and the overall performance also improved. To gauge whether human expertise in machine learning was important to our process, we then asked 22 human participants without any particular background in machine learning to modify the reasoning logic (in Appx. Sec. A). We

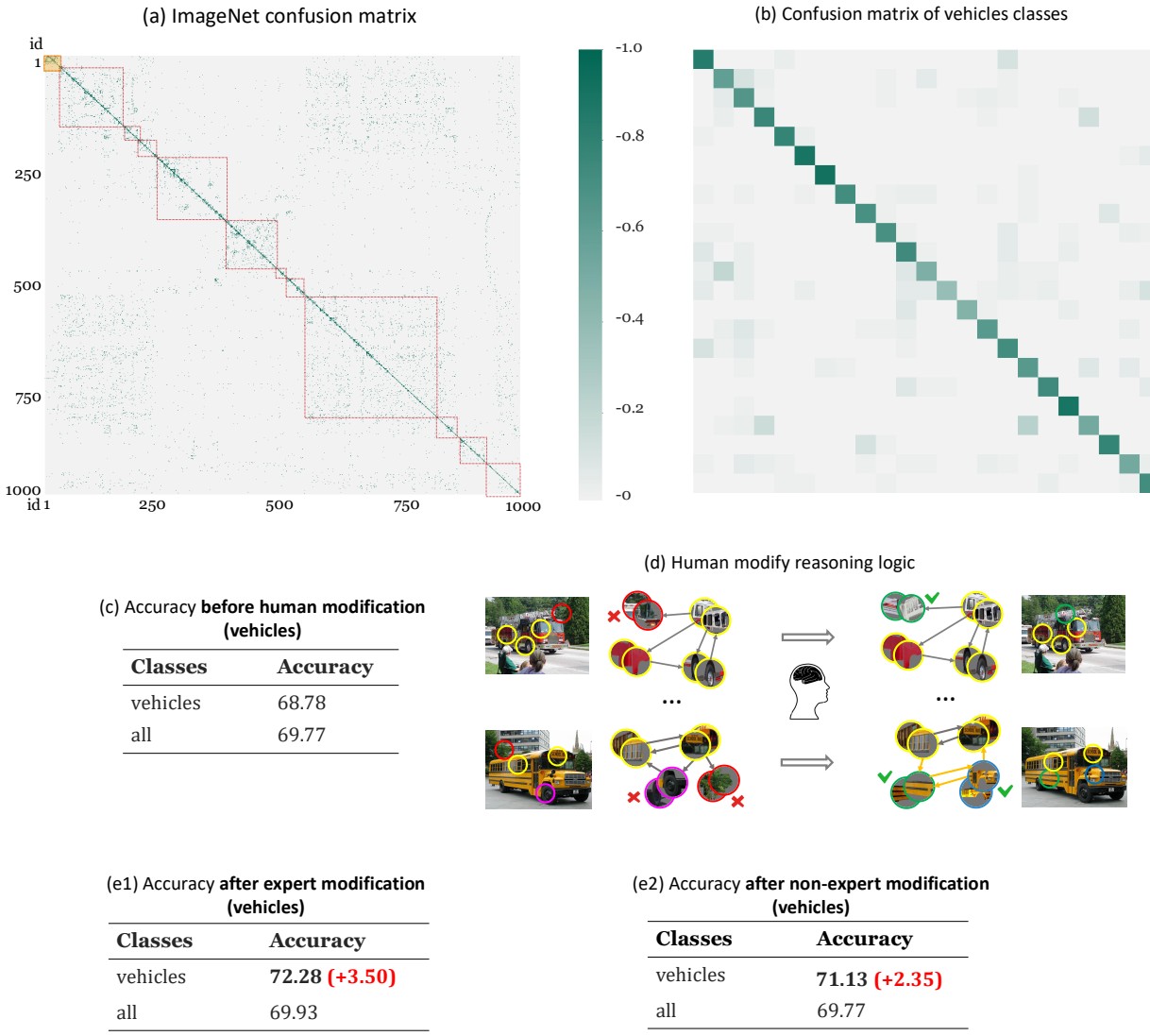

Figure 7: Both expert and non-expert opinions can improve a network's performance with HNI. (a) Confusion matrix of a 1,000-class original GoogleNet image classification network trained on ImageNet. We consider the super-class of vehicles (b), with original accuracy over these 68.78% (c). For each class, the network-to-human pass was used to show the reasoning logic of the original network as a c-SCG to a human operator (d). The operator spotted and corrected any reasoning errors of the network. The human-to-network pass then distilled the human-modified logic back to the original network with the help of Graph neural network and partial knowledge distillation. Performance was improved on the vehicle classes, without degradation of all classes, demonstrating how both experts (e1) and non-expert (e2) could use their own knowledge to correct reasoning errors of the network and improve network accuracy.

designed a web interface that participants used to visualize and modify the network's reasoning logic. As shown in Fig. 7(e2), we can find similar improved results on classes of interest (vehicles).

## 4.3   Zero-shot learning: Human teach network to learn new object through HNI

We introduce a novel zero-shot learning pipeline with the proposed Human-Network-Interface (HNI). The high-level idea is that understanding a new object can be represented as a new class-specific c-SCG, consisting of visual concepts (nodes) and concept relationships (edges). Our HNI allows humans to design new c-SCGs

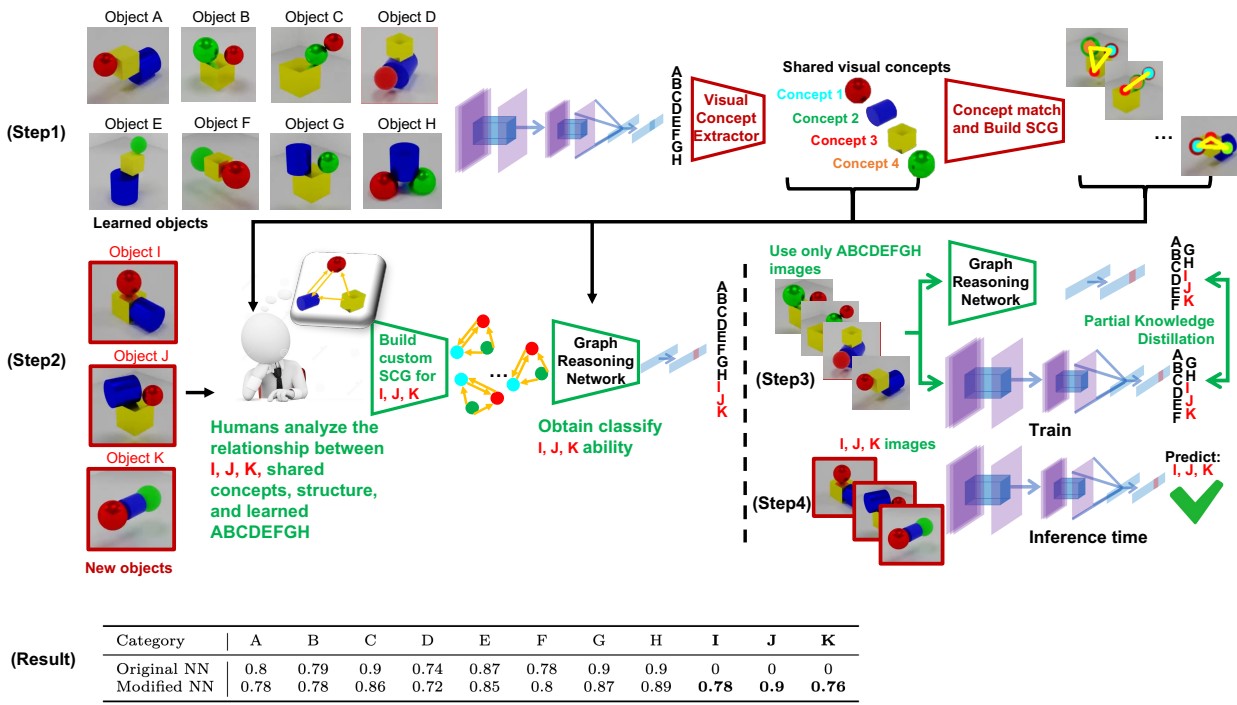

Figure 8: Zero-shot learning: Human users teach network to learn to encode new objects with HNI. Sec. 4.3 provide explanation for each step. Bottom results shows the performance of Zero-shot learning with HNI. The original ResNet-18 network (pretrained on ImageNet) trained with images of objects A-H cannot identify new objects I, J, K in the test set. Humans can teach the ResNet-18 to encode and recognize new objects I, J, K wih HNI.

for new object categories, using existing primitive visual concepts (nodes) or relationships (edges) discovered from other classes. The new c-SCGs can then be distilled back to the original network, thereby guiding the original network to encode new object categories (i.e., zero-shot learning). While learning to recognize new classes, the original network will not "forget" the old classes. In our experiment, the network first learned 8 objects A, B, C, D, E, F, G and H using backpropagation and 300 training images and 216 test images. Then, new objects I, J, K were defined by humans and have only 216 test images each. In this experiment, I, J, and K had novel overall shapes but used the same parts as A-G (ses Discussion). Fig. 8 shows the detailed workflow:

**Step 1: network-to-Human:** We train a classifier for 8 objects A to H (Fig. 8 Step 1). We use VCE to discover the visual concepts and find the shared concepts. We match the shared visual concepts from training images and form image-level SCGs (I-SCG) for all training images of objects A to H.

**Step 2: Human-to-network: Building new c-SCGs for new objects I, J, K and training new GRN:** (Fig. 8 Step 2) It is straightforward for humans to learn about a new class as they can relate the patterns and components on the new class to those they have seen in the past. We try to implement a similar mechanism here in describing the new class with SCG to GRN. We construct novel I-SCG training instances with visual concepts and relationship from learned classes, in an automated fashion. For instance, to form a I-SCG for new object I, we know some of its components are overlapping with objects A to H. Hence we randomly sample one I-SCG from object A and use its concept 1 as the node of concept 1 in I's I-SCG. Similarly, we obtain I's nodes of concept 2 and 3 by randomly sampling I-SCGs of objects E and G. To construct I-SCG edges for object I, we sample I-SCGs of D and form the edges of I based on their similar structures. Building I-SCG for new objects J and K are similar. We form a new I-SCG training set by adding the novel I-SCGs of I, J, K into the original training set and then train a GRN that can classify objects A to H and I, J, K.

**Step 3: Transfer knowledge from GRN back to original network:** (Fig. 8 Step 3) we use knowledge distillation to transfer the knowledge about new object I, J, K from GRN back to the original network. In this process, we *only* use the images of A to H as the training set, and only use the soft label form GRN without any hard label to avoid bias toward old classes.

**Step 4: Network learns the knowledge to encode new objects** without forgetting the knowledge about old classes (Fig. 8 Step 2). Fig. 8 bottom result shows the performance of zero-shot learning with our HNI. We will make the OBJECT dataset which we created and used here public (more details in Appx. Sec. A).

### 4.4 More edges (concept relationships) modification experiments

We conduct an edges (concept relationships) modification experiment on the iLab-20M Borji et al. (2016) dataset, which has a controllable pose to introduce bias on concept relationships. iLab-20M contains images of toy vehicles placed on a turntable using 11 cameras at different viewpoints. We tailor a subset of iLab-20M to train a three-class vehicle classifier with ResNet-18: bus, military, and tank. In the training set, each class has 400 images. We manually introduce biases in pose of each class: all buses are with pose 1, all military vehicles are with pose 2 and all tanks are with pose 3 (Fig. 9 (a)). We construct an unbiased test set where each kind of vehicle has all the 3 poses.

After training, we use the network-to-human path to reveal the reasoning logic of the original network to humans by visualizing the c-SCG and explaining the reasoning logic on misclassified images using I-SCG. Human users found that important concepts in c-SCG are mostly in the foreground and are consistent with human user's own understanding of the class of interest. Explanation of incorrectly predicted images also shows similar results that most of the detected visual concepts had a positive contribution to the correct class. Thus, no node modification was needed. However, the structural relationship between concepts contributed mostly negatively, which caused incorrect predictions (Fig. 9 (b)) (we use similar positive/negative contribution analysis in Ge et al. (2021)). To eliminate the unstable/independent concept relationship, human users delete all edges in SCG and train a GRN with the new c-SCG. The new GRN focuses only on visual concepts during decision-making, while it does not pay attention to the relationship between them. Although, in general, deleting all edges may be too extreme, in this case it is one reasonable way to correct the problem as most errors come from pose bias in the training set, and edge features in SCG rely heavily on the pose of target objects in this case. After the modification, we use partial knowledge distillation to modify the decision logic of network. The final performance of the modified network shows improvement compared with the original network (Fig. 9(c)), demonstrating that HNI can help humans to improve original network's performance with an intuitive interface and effective mechanism in modifying the concept relationship and exchanging knowledge. In other applications, humans may modify nodes and edges simultaneously to improve the performance of network. To evaluate the user feedback of HNI. We conducted a human user study (in Appx. Sec. A) with responses from ML practitioners and students.

## 5 Conclusion

We showed that HNI addresses three key challenges in human and Neural Network (network) knowledge exchange. The first is "what is the language to communicate". The interface needs a "language" to represent the reasoning logic and knowledge in a manner comprehensible to both humans and networks. We propose using Structural Concept Graphs (SCGs) to represent the network's reasoning logic in a format that is understandable by humans. We demonstrate that the network-to-Human path employs SCGs to offer local and global explanations of the network's reasoning logic, making it accessible and clear to human users: (1) Class-specific Structural Concept Graphs (c-SCGs) elucidate the network's understanding of each class by highlighting the critical visual concepts employed during decision-making and the relationships between them (global explanation). The local explanation (image-wise) addresses "Why" and "Why not" questions, revealing the network's reasoning logic behind its decisions. Furthermore, the Network-to-Human path can impart new knowledge to humans by demonstrating the network's reasoning process.

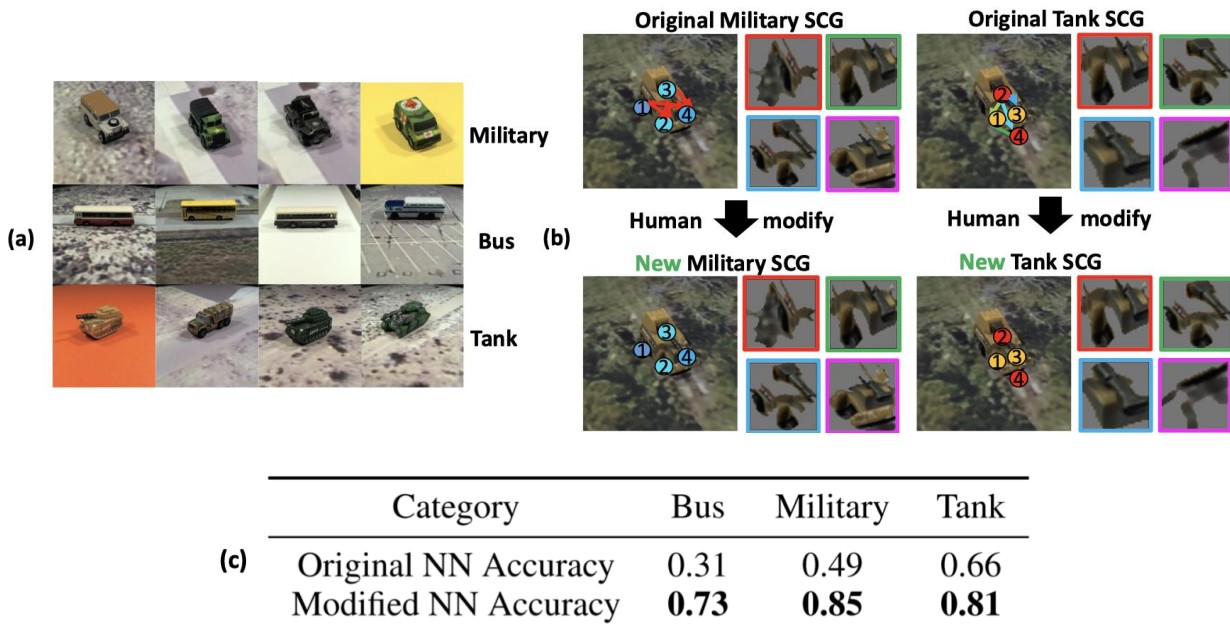

Figure 9: Example of edges (concepts relationship) modification. (a) biased iLab dataset. (b) Human user can remove incorrect edges to guide the model to ignore irrelevant concept relationships introduced by dataset bias. (c) iLab-20M three class classifier performance.

The second challenge is "how can humans share their knowledge with the network". Unlike traditional interaction methods where humans participate in the network's training process by gathering new data and retraining the network, our Human-to-Network path enables humans to directly modify the c-SCG with their knowledge, thereby altering the network's reasoning logic. We demonstrate that using Graph Reasoning Network and Partial knowledge distillation, humans could improve the performance of the original network for a local logic of interest, without degrading performance on the unmodified classes. Notably, human modification of c-SCG is highly efficient, because one only needs to inspect one graph with a few nodes and edges for each class. This efficiency significantly reduces time and monetary costs compared to traditional methods, and, importantly, it also offers the intuitive interpretability advantage of our HNI.

The Third challenge is "how can humans teach networks new knowledge". For instance, humans may want to teach networks about novel classes. Compared with mainstream zero-shot learning settings (Geng et al. (2020); Fu et al. (2018); Xian et al. (2018)), which provide attribute descriptions for images (e.g., stripes, horse-like shape, big four-legged animal), our method considers a more general setting and we make no assumption on the availability of attribute labels. Instead, we rely on the unsupervised mining of primitive concepts from the training dataset, without requiring attribute or concept labels. By combining different subsets of these learned primitive concepts and varying the structural relationships between them, we can use GCN to represent novel classes and eventually guide the network to learn to encode them by HNI. While our method has limitations, particularly when a new class cannot be easily represented by the learned primitive visual concepts, it offers flexibility in defining new structural or spatial relationships based on learned relationships. We posit that using larger datasets with more classes than just the A-G classes in our zero-shot experiment will mitigate this limitation, as the larger number of classes may provide a richer pool of visual concepts for humans to choose from when defining new classes. This strength makes our method more extensible and capable of encoding novel objects with fewer assumptions, not being confined to a given list of attributes for describing relationships or structures of components and parts of novel objects.

**Broader Impact Statement**

HNI allows humans and network to understand each other using SCG as a "language". A human can directly modify network with human prior knowledge. It can also help us "teach" network to learn new objects. This can have a positive impact on: (1) We can teach network not to learn some knowledge we do not want it to learn, e.g., not learn some social bias; (2) we can help network learn something it has never seen before. This will be useful when we lack the data. We can use our prior knowledge to build SCG and help network to learn the knowledge of this object.

However, those who have bad intentions may be able to figure out a way to use it maliciously. In our case, one might find a way to teach network something unsuitable. There are more and more scenarios using network to help humans to make decisions. If we teach network unsuitable knowledge, it may make wrong decisions, but that could be done through standard training as well.

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

## A  Appendix

## Human-in-the-Loop User Study

We conducted a user study with responses from 22 students giving us the following data. We utilized a web-based interface to display the original reasoning of the network, which included the c-SCG with the top 4 visual concepts, as well as an additional 21 top-ranked visual concepts. The study protocol was reviewed and approved by our Institutional Review Board (IRB).

For practitioners: (1) they are volunteers to join the human-in-the-loop User Study and are compensated USD 10 for their participation. (2) the age range is 18 to 30 years old.

Potential participants who may have heard of this study from their friends will visit the study's online survey.

They will first see several practice questions on teaching them how to choose good concepts and edges (Fig. 10). Then participants will see a following SCG graphs (Fig. 11(a)), each of them represents an individual class. For instance, as Fig. 11(b)) shows, participants were first asked to select the best 4 visual concepts from a total pool of 25. They were informed that, in some cases, the original 4 concepts chosen by the network might already be the best options, but in other instances, they might include errors. After selecting the concepts, participants were shown the edges in the original c-SCG and asked if they wanted to modify them to alter the importance of relationships between concepts, as illustrated in Fig. 11(c)).

## More Details of OBJECT Dataset

OBJECT is a computer-generated 3D object image dataset using Blender Community (2018) as the rendering engine. Each image contains a 3D main object which consists of multiple basic object parts: ball, cylinder cubic, etc. Each main object is rendered using 4 independent generating attributes: object size, background color, rotation angle, sub-object material. The image size is 512 x 512. The first version of the dataset contains 13 different main objects and each one has more than 400 images. We also publicly distribute the source code, which allows one to render new data with custom main objects. The domain of attributes and resolution depends on the needed dataset size. Fig. 12 shows an example that also illustrates the workflow of

**Section 2. Practice Questions**
(1) In this section, you'll be presented with **images depicting a specific class of object and images of various components** extracted from this object. Your task is determining **whether a given component is an essential concept for recognizing this object**. Please note the images of the components may not be from the exact instance of the objects presented in the question.

(2) To facilitate a better understanding, we'll use a **yellow circle** in the second row to indicate the origin of each component within the object.

(3) Feel free to **click** on the images for an enlarged view when you need a closer examination.

**Section 2.1. Example question: Select the components**

*2. Three images from the ambulance class.

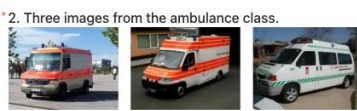

**(1) Example of a good concept.** This concept features elements that closely resemble the car door of an ambulance, including the unique texture characteristic of this type of vehicle. This specific feature is vital for accurately identifying the object as an 'ambulance'.

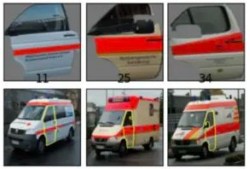

**(2) Example of a bad concept.** While the image of the wheels is a component of the ambulance, this feature is not unique to ambulances and resembles the wheels of many other types of vehicles. Therefore, it does not significantly contribute to differentiating an ambulance from other vehicles, making it less important for identification.

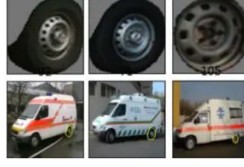

◯ I have understood this case.

**Section 2.2. Example question: Select the edges**

*5. Three images from the ambulance class.

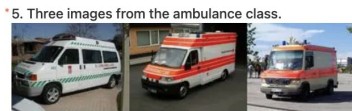

AI Interpretations.

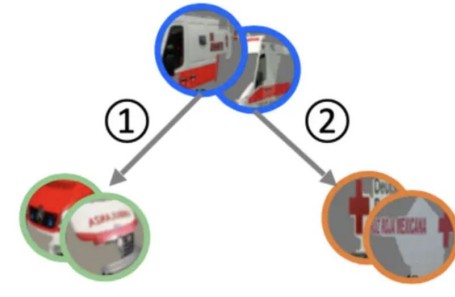

**Node**: A concept can be a node when it is an important feature to recognize this object.

**Edge**: An edge from node A to B indicates that concept A is important to B. This implies that knowledge about A is essential to infer information about B, such as spatial position or semantic relationship.

**Question: Which edge(s) would you recommend deleting?**

**Answer:**

**(a) Edge ① should be kept.** Once we know the location of the "left side of the cars", we could possibly know where their "heads" are. This signifies a strong spatial correlation between these two concepts of the cars.

**(b) Edge ② ought to be removed.** This edge connects the "left side of the cars" to the "red crosses" concept. However, the spatial relationship of the red cross symbol is not limited to the left side of the vehicles; it could also be located on the front or back. Therefore, the spatial relationship between these two concepts does not hold consistently in reality.

◯ I have understood this case.

Figure 10: Practice questions of questionnaire. The participants will first see several practice questions on teaching them how to choose good concepts and edges.

new object images generation: User can select the basic objects to form the main object, we will output the corresponding three-view drawing helping people to make sure the main object configuration and parameters are correct. After the user confirms the main object configuration, we can automatically synthesize over 400 images for user defined main object with different rendering attributes mentioned above. The output datasets will contain all possible combinations of the attributes. Our primary motivation for creating the OBJECT datasets is that it allows fast testing and idea iteration, on zero-shot learning and 3D object recognition.

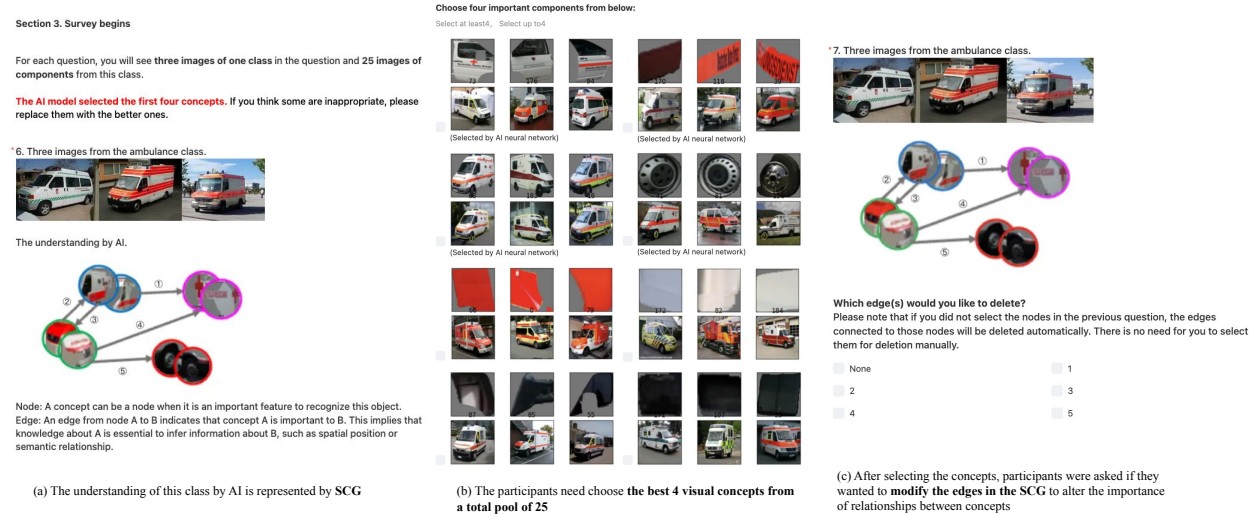

(a) The understanding of this class by AI is represented by **SCG**

(b) The participants need choose **the best 4 visual concepts from a total pool of 25**

(c) After selecting the concepts, participants were asked if they wanted to **modify the edges in the SCG** to alter the importance of relationships between concepts

Figure 11: The experiment will proceed with a series of questions, each asking to choose the correct answer for a given class. For instance, this is a set of example questions for the Ambulance class. The participants are shown with the SCG graph which represents the understanding of this class by AI model (a). Then the participants are asked to corrected the original selection of visual concepts (b) and edges (c).

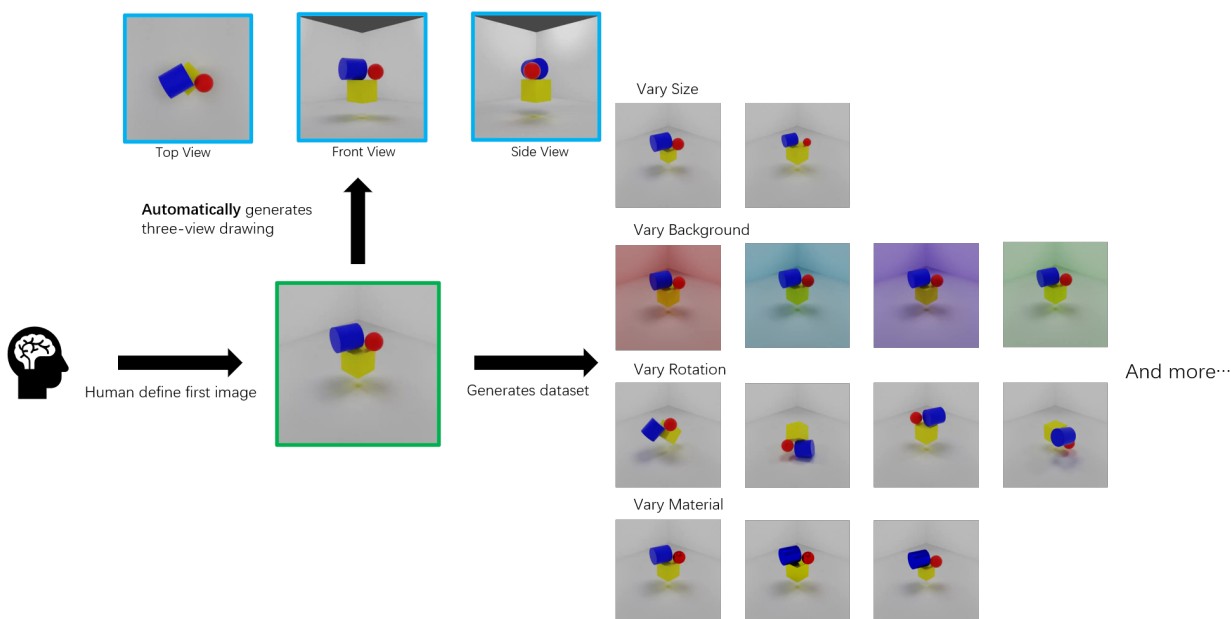

Figure 12: Overview of OBJECT dataset

