# OpenReview forum: "A Graphical Framework for Knowledge Exchange between Humans and Neural Networks"
_TMLR — Rejected by TMLR_

### Review · Reviewer_pF5j · 2025-12-03

**Summary Of Contributions:**

Summary \
This paper suggests Human-Neural Network Interface (HNI), a new framework for interpretable and human-in-the-loop machine learning. HNI extracts visual concepts from training data and constructs a concept graph for each class which enables humans to understand its reasoning logic. Then, humans can modify the incorrect logic of the network by deleting the edges or changing the nodes of the graphs. This human knowledge is transferred to the networks with partial knowledge distillation. The experiments are conducted on various image classification tasks to show that HNI improves the performance with human interaction.

Strengths
- The paper introduces a new framework HNI which enables humans and the network to exchange knowledge.
- The paper creates a new dataset (OBJECT) which the researchers would be interested in using if released publicly.
- HNI improves performance on various image classification tasks.

Weakness
- The paper is rather verbose. Lots of the parts could be shortened or moved to the appendix (e.g., about 2 pages of related work section which feels like simply listing all the prior works rather than providing a meaningful message). In this regard, I am not sure if the long submission is an appropriate choice.
- The paper argues the efficiency of the introduced mechanism but is missing a related analysis.
- The paper is missing a comparison with important prior works [1].

**Additional Comments:**

References \
[1] Concept Bottleneck Models, Koh et al, ICML 2020 \
[2] Concept Embedding Models: Beyond the Accuracy-Explainability Trade-Off, Zarlenga et al., NeurIPS 2022

**Audience:**

Yes

**Audience Explanation:**

There is a growing interest in the field of interpretable machine learning / human-in-the-loop machine learning. The paper introduces a new framework for this area which researchers would be interested in the architecture, mechanism, and the experimental results. The paper also provides a new dataset called OBJECT which researchers in the related area would be interested in if released publicly.

**Broader Impact Concerns:**

The paper properly addresses ethical concerns in the Broader Impact Statement section.

**Claims And Evidence:**

No

**Claims Explanation:**

First, the paper argues that ``human users and network can more **efficiently** communicate between each other’’ using HNI, but this claim is not clearly supported for the following reasons.
- The computational cost of extracting concepts, building concept graphs, and concept matching is not provided. This cost can be large when the number of classes or the number of data for each class is large.
- It would be time-consuming for humans to modify the nodes of concept graphs since they have to check one by one if each item in the concept pool (where the number of items should be large to ensure that there is an appropriate substitute) is an appropriate substitute for the deleted node. For this procedure to be efficient, the framework should incorporate a mechanism which first automatically selects a small number of possible candidates followed by human choosing a substitute among these candidates.
- The paper should provide a theoretical cost analysis or actual runtime analysis (compared to prior works, e.g., intervention procedure in [1]) to support the efficiency of HNI.

Next, in Figure 7, the paper argues that non-expert opinions can also improve a network’s performance. Here, the expert is defined as a person with knowledge in machine learning. However, in practical scenarios, the expert should be the one with knowledge in target tasks, e.g., a person with knowledge of vehicles for this example. A prior work [2] has shown that the performance of the network decreases significantly when human interventions are incorrect (which can frequently arise from non-experts).

Finally, the paper argues that the performance of concept matching is not sensitive to the threshold $t$ but is missing a related analysis.

**Requested Changes:**

(major) Analysis on the efficiency: The paper should provide an analysis of the computational cost or runtime to argue that the provided mechanism enables efficient communication between humans and the network.

(major) Comparison with [1]: The paper is missing a comparison with closely related work [1] which provides the intervention mechanism to modify the reasoning logic of the neural network. It is recommended that the authors compare HNI with [1] in terms of design, performance, and computational efficiency.

(minor) Consistent naming: It would be better to name the subsections in a consistent manner. For example, Sec 3.2.1 would change to ``Modifying c-SCG’’ to keep the name in a consistent style as in Sec 3.2.2.

(minor) Sensitivity analysis: Experiments on the sensitivity analysis of threshold $t$ should be included to support your claim (“the performance is not sensitive to $t$”).

---

### Review · Reviewer_EVze · 2025-12-09

**Summary Of Contributions:**

The paper introduces a graphical “human–neural network interface” (HNI) for image classification, built on top of structural concept graphs. Starting from a trained CNN, the authors first derive class-level structural concept graphs (c-SCGs), where each node represents a visual concept extracted from intermediate feature maps, and edges encode spatial or relational structure between these concepts. They then train a graph reasoning network (GRN) to take these graphs as input and reproduce the original network’s predictions. On top of this, humans can directly interact with the c-SCGs: they can add, delete or substitute nodes to tweak the set of visual concepts, and adjust edges to change how concepts relate to each other. These edits are interpreted as injecting human prior knowledge and fixing spurious correlations or missing factors in the model’s reasoning. The updated graphs are fed through the GRN to obtain pseudo-labels, and a partial knowledge distillation scheme is used to transfer this “human-edited” decision logic back into the original CNN. The authors also explore a zero-shot scenario on a synthetic OBJECT dataset, where new classes are formed by recombining concepts and relational structures drawn from existing classes.

Key strengths:
1) The paper puts together a coherent workflow that links three stages: structured explanations via SCG-based concept graphs, human edits on those graphs, and then distilling the updated behavior back into the original CNN.

2) Human interaction happens at the class-level concept graph rather than at the level of individual images, so the interface is relatively lightweight and practical for human-in-the-loop use.

3) The partial knowledge distillation design explicitly handles the case where only some classes are edited while others remain untouched, and this mechanism fits fairly naturally into the overall framework.

4) On several medium-scale datasets and ImageNet subsets, the experiments suggest that tweaking a small number of problematic classes can noticeably improve performance for those classes.

Key weaknesses:
1) Compared with earlier work on SCG/VRX-style structural explanations and knowledge-graph–based methods, the conceptual step forward feels modest, and the paper does not fully convince the reader that the gap to prior art is large.

2) Most comparisons are between the original model and the proposed HNI framework. There is no direct comparison against simpler human-in-the-loop baselines (for instance, targeted fine-tuning using a few carefully chosen hard examples), so it is difficult to judge whether the extra machinery is really needed.

3) The zero-shot experiments are carried out on a simplified synthetic OBJECT dataset, without comparisons to standard ZSL baselines. As a result, it is hard to gauge how strong or general this component actually is.

4) Several technical pieces are described only at a high level (e.g., details of the graph convolutions, thresholds for concept matching, and the robustness of partial KD to hyperparameter choices), which hurts reproducibility and makes it harder to fully trust the method.

**Audience:**

Yes

**Audience Explanation:**

The paper lies at the intersection of several active research areas:
1) Model interpretability and explanation via structured concept graphs
2) Human-in-the-loop machine learning, where humans can directly manipulate an intermediate representation rather than only labels
3) Knowledge distillation and knowledge injection, with a focus on blending human-modified reasoning with an existing model.
4) Zero-shot and low-data generalization, although this aspect is currently demonstrated in a limited setting.

The idea of giving humans a graphical, concept-level interface to inspect and correct a model’s reasoning, and then distilling these corrections back into the model, is conceptually appealing. Even if the empirical evaluation could be strengthened, many readers interested in interpretability, interactive ML, and structural representations are likely to find the framework and empirical observations thought-provoking. The work can serve as a reference or starting point for further exploration of human-editable intermediate representations and structured knowledge integration into deep models, and therefore at least some individuals in the TMLR audience would be interested in the findings of this paper

**Claims And Evidence:**

Yes

**Claims Explanation:**

In my view, the core claims are reasonably supported, but some of the broader or more ambitious claims are only partially backed by the evidence presented.

The paper provides qualitative and quantitative evidence that he proposed structural concept graphs and graph reasoning network can approximate the original model’s predictions and expose class-level reasoning patterns. Human edits to these concept graphs (removing spurious concepts, adding more discriminative ones, adjusting relations) can lead to measurable performance improvements on selected problematic classes, while not significantly harming overall accuracy on the remaining classes.

**Requested Changes:**

1. Stronger positioning and comparison to closely related work. Please clearly articulate how the proposed HNI framework differs from and improves upon prior SCG/VRX-style methods and knowledge-graph–based approaches. Ideally, include at least qualitative or small-scale quantitative comparisons that highlight the distinct benefits of the proposed pipeline over these baselines, beyond simply allowing graph editing. (Critical)

2. Please add baselines for simpler human-in-the-loop interventions. For the class-specific improvements, i'd like to see comparison between the HNI pipeline and more straightforward strategies such as: (i) targeted fine-tuning on a small set of hard examples for the problematic classes, and/or (ii) simple reweighting or margin adjustments based on human feedback. (Critical)

3. Strengthen or temper the zero-shot learning claims. Either (a) extend the zero-shot evaluation to at least one standard image-based ZSL dataset and include comparisons with basic ZSL baselines, or (b) explicitly frame the OBJECT results as a limited proof-of-concept and soften claims about zero-shot capabilities in the main text. In either case, provide more details on how the new classes are constructed from existing concepts and how well this approach is expected to scale to more realistic settings. (Critical)

4. Clarify technical details of the GRN and partial KD, and provide some sensitivity analysis. The authors need to give explicit update equations for the graph convolution layers, including how node and edge attributes are used and combined. They also need to explain the rationale behind the partial KD probability reallocation and the choice of temperature and loss weights more carefully. (critical)

5. Provide more detailed and quantitative information about human effort and variability. You may quantify the actual time spent per class and per user, the number and expertise of annotators, and the amount of variation in edits across annotators. (not critical)

6. More ablation studies. A. editing only nodes vs only edges vs both; B. using HNI only for explanation vs explanation + distillation; C. variants of concept matching strategies. (critical)

7. Expand discussion of limitations and future directions. Please acknowledge more explicitly that the current instantiation is tailored to CNN-based image classification and that extending HNI to other architectures such as vision transformers and tasks such as detection and segmentation is non-trivial.  (not critical)

---

### Review · Reviewer_MWEn · 2026-01-12

**Summary Of Contributions:**

This paper introduces a new pipeline for HNI (Human-Neural Network Interface), enabling bidirectional knowledge exchange between humans and neural networks. HNI incorporates network reasoning using class-specific structural concept graphs (c-SCGs), which humans can inspect and modify to correct or extend the network’s knowledge. Human modifications are distilled back into the network via a Graph Reasoning Network, improving performance and enabling zero-shot learning. The approach is computationally efficient and is validated on large-scale image classification and zero-shot tasks.

**Strengths:**
- The HNI pipeline is well structured and aligns closely with the goals of interpretability and controllable model revision. The idea looks interesting.
- The use of compact, graph-based concept representations offers a suitable high-level abstraction that supports intuitive human interaction and systematic model refinement.

**Weaknesses**

The most concerning part is that Figures 4 and 5 are visually identical but have different captions, creating redundancy and ambiguity regarding their distinct roles. The confusion matrix visualization reports values in the range $[-1, 0]$ without explanation, making interpretation difficult. Finally, several citations are missing, including references to HuntGPT[1] and foundational work on knowledge distillation[2], which weakens the scholarly grounding of the introduction.

In Figure 1, visual concepts appear connected for only one example image, while the remaining examples do not show such connections. This is confusing, as c-SCGs are not expected to appear on the left-hand side immediately after concept extraction. The rationale behind this visualization choice is unclear and should be clarified.

The loss formulation eq(1) introduces both $\alpha$ and $\beta$, resulting in unnecessary degrees of freedom without sufficient justification. Moreover, no ablation study is provided to disentangle the contributions of the hard and soft objectives, which obscures the methodological impact of each component.

While increased human involvement is presented as a strength, the paper does not adequately discuss its associated costs or risks. In particular, there is no quantification of human effort (e.g., time required to modify c-SCGs), nor is there an analysis of potential human-induced biases. All reported experiments show consistent performance improvements, raising concerns about whether negative or failure cases exist when human interventions are suboptimal or biased. I was wondering if the authors can share when this pipeline may fail with the human bias involved?

The experimental evaluation relies primarily on accuracy, which is insufficient for both classification and multiple-choice tasks. Metrics such as precision, recall, and F1-score are not reported. Additionally, results are presented without error bars or variance estimates, and no multiple-seed experiments are conducted.

Although the HNI framework is claimed to be network-agnostic, all experiments appear to use a single backbone (GoogLeNet), limiting evidence for generality. Furthermore, no sensitivity analysis is provided for key hyperparameters, including the number of concepts per c-SCG ($k=4$) and the distillation temperatures $T_{T_1}$ and $T_{T_2}$.

Several minor presentation issues undermine the readability of the paper and occasionally lead to confusion about terminology and methodology. First, Figure 9(c) presents results in a non-LaTeX table format, which is inconsistent with Figure 8 and detracts from clarity. Second, terminology is inconsistent: **Graph Reasoning Network** appears in the introduction without an explicit abbreviation, while later sections use $GRN$, suggesting an undefined new term. Similarly, the term `CNN` is used without clarification and appears to implicitly refer to GoogLeNet, which may confuse readers.

**References**

[1] Ali, T., & Kostakos, P. (2023). Huntgpt: Integrating machine learning-based anomaly detection and explainable ai with large language models (llms). arXiv preprint arXiv:2309.16021. \
[2] Hinton, G., Vinyals, O., & Dean, J. (2015). Distilling the knowledge in a neural network. arXiv preprint arXiv:1503.02531.

**Audience:**

Yes

**Audience Explanation:**

It is interesting that this new HNI pipeline leverages human modifications of c-SCGs and employs a Graph Neural Network to transfer this knowledge back to the neural network. This approach will likely be of interest to human-centered ML researchers and those working on explainable AI.

**Claims And Evidence:**

No

**Claims Explanation:**

There are four key claims in this paper. The main issues that I have seen so far lie in the experiment part. This makes me doubt that if there are enough evidence to validate this HNI pipeline. The details are listed in the Requested Changes section. The claim that HNI establishes a "novel pipeline for zero-shot learning" is primarily supported by experiments on a custom, synthetic OBJECT dataset. While this proves the mechanism works in a controlled environment, it lacks evidence on standard real-world zero-shot learning benchmarks, which makes the "clear and convincing" evidence for generalizability somewhat weak. Another claim "HNI modifies the original model’s parameters with minimal overhead. This design ensures high compute efficiency and avoids the large-scale training or storage costs". However, the paper does not provide a table or quantitative summary to substantiate these efficiency claims.

**Requested Changes:**

This list of changes is associated with the weaknesses above.
- Resolve and clarify figure-related issues, including converting Figure 9(c) to a LaTeX table, merging or clearly differentiating Figures 4 and 5, and clarifying the visualization logic in Figure 1. (minor)
- Strengthen experimental validity by reporting variance (error bars), running experiments with multiple random seeds, and including evaluation metrics beyond accuracy (e.g., precision, recall, F1 for multi-class classification tasks). (major)
- Add missing citations, including references to HuntGPT and foundational work on knowledge distillation (e.g., Hinton et al.). (major)
- Simplify and better justify the loss formulation, for example by using a single weighting parameter and providing ablation studies on $\alpha$ and $\beta$. (major)
- Conduct sensitivity analyses not only for the number of concepts per c-SCG and the distillation temperatures, but also for the threshold $t$ in sec. 4.1.2 concept matching. (major)
- Evaluate the generality of the HNI framework by testing it with additional backbones (e.g., ResNet[3] and ResNeXt[4]). (Major)
- Explicitly discuss human effort and bias, including time cost measurements and potential failure cases. (minor)
- Clarify the interpretation and value range of the confusion matrix. (minor)
- Fix terminology and notation inconsistencies by introducing and consistently using abbreviations (e.g., GRN) and clarifying what “CNN” refers to in this sentence "As a Graph Neural Network (GNN) based network, GRN takes a graph as input but has a similar decision as the original network (CNN)." (minor).

**References**

[3] He, K., Zhang, X., Ren, S., & Sun, J. (2016). Deep residual learning for image recognition. In Proceedings of the IEEE conference on computer vision and pattern recognition (pp. 770-778). \
[4] Xie, S., Girshick, R., Dollár, P., Tu, Z., & He, K. (2017). Aggregated residual transformations for deep neural networks. In Proceedings of the IEEE conference on computer vision and pattern recognition (pp. 1492-1500).

---

### Decision · Action_Editor_nQRf · 2026-02-24

**Recommendation:** Reject

**Additional Comments:**

The reviewers share some common concerns.
For example, the claimed efficiency of the proposed framework is not empirically validated (Reviewer MWEn and pF5j).
Several key references and baselines (e.g., the ones mentioned by Reviewer EVze and pF5j) are missing.
More ablation study and sensitivity analysis are required (Reviewer MWEn, EVze, and pF5j).
Some technical details need more clarification (Reviewer MWEn amd EVze).

Overall, given that no rebuttal/discussion from the author is received, I recommend rejection and encourage the authors to improve the manuscript based on the feedback.

**Audience:**

Yes

**Audience Explanation:**

The core contribution, i.e., a graphical-form interface for humans to 1) inspect and modify a deep learning model’s reasoning and 2) merge and distill corrections back into the model, is conceptually appealing.
This would be interesting to researchers who work in the field of interpretable machine learning, human-in-the-loop machine learning, knowledge distillation, and so on.
Furthermore, the paper also provides a new dataset called OBJECT which would benefit the community if released publicly.

**Claims And Evidence:**

No

**Claims Explanation:**

The reviewers have identified a few claims that are not well supported.

1. ***HNI modifies the original model’s parameters with minimal overhead. This design ensures high compute efficiency and avoids the large-scale training or storage costs*** is not supported by any experiments that investigate the efficiency [Reviewer MWEn, pF5j]
2. ***HNI establishes a novel pipeline for zero-shot learning*** is only validated on synthetic OBJECT dataset. [Reviewer MWEn]
3. ***The performance of concept matching is not sensitive to the threshold $t$*** is not supported by any analysis. [Reviewer MWEn, pF5j]

**Resubmission Of Major Revision:**

The authors may consider submitting a major revision at a later time.